# Targeted Therapies in Rare Brain Tumours

**DOI:** 10.3390/ijms22157949

**Published:** 2021-07-26

**Authors:** Francesco Bruno, Alessia Pellerino, Luca Bertero, Riccardo Soffietti, Roberta Rudà

**Affiliations:** 1Department Neuro-Oncology, University and City of Health and Science, 10126 Turin, Italy; alessia.pellerino@unito.it (A.P.); riccardo.soffietti@unito.it (R.S.); rudarob@hotmail.com (R.R.); 2Pathology Unit, Department of Medical Sciences, University and City of Health and Science, 10126 Turin, Italy; luca.bertero@live.it

**Keywords:** rare brain tumours, targeted therapies, molecular neuro-oncology

## Abstract

Rare central nervous system (CNS) tumours represent a unique challenge. Given the difficulty of conducting dedicated clinical trials, there is a lack of therapies for these tumours supported by high quality evidence, and knowledge regarding the impact of standard treatments (i.e., surgery, radiotherapy or chemotherapy) is commonly based on retrospective studies. Recently, new molecular techniques have led to the discovery of actionable molecular alterations. The aim of this article is to review recent progress in the molecular understanding of and therapeutic options for rare brain tumours, both in children and adults. We will discuss options such as targeting the mechanistic target of rapamycin (mTOR) pathway in subependymal giant cells astrocytomas (SEGAs) of tuberous sclerosis and BRAF V600E mutation in rare glial (pleomorphic xanthoastrocytomas) or glioneuronal (gangliogliomas) tumours, which are a model of how specific molecular treatments can also favourably impact neurological symptoms (such as seizures) and quality of life. Moreover, we will discuss initial experiences in targeting new molecular alterations in gliomas, such as isocitrate dehydrogenase (IDH) mutations and neurotrophic tyrosine receptor kinase (NTRK) fusions, and in medulloblastomas such as the sonic hedgehog (SHH) pathway.

## 1. Introduction

According to the last Central Brain Tumour Registry of the United States (CBTRUS) Statistical report, the overall annual incidence of primary central nervous system (CNS) tumours diagnosed in the US between 2013 and 2017 is 23.79 out of 100,000 people per year [1]. This incidence, which may seem quite high per se, is the sum of separate and diverse conditions that are rare and differ in terms of histology, molecular characteristics, clinical features, and outcome. Except for tumours arising from meninges, the majority of CNS tumours may be defined as ‘rare’, according to the RARECARE definition of <6 cases/100,000/year [2]: for example, neuroepithelial tumours (which comprise gliomas, ependymomas, and glioneuronal tumours) account for approximately 7 cases/100,000/year, half of which are represented by glioblastoma (approximately 3 cases/100,000/year), and the rest by all other entities [1]. In the last two decades, knowledge of molecular mechanisms that promote tumourigenesis has impressively evolved. As a result, entire classes of tumours that were previously considered as unique histological categories have been scattered into smaller subgroups identified by specific molecular features with diagnostic and/or prognostic importance. This has allowed for a greater understanding of brain tumours’ biology and improvement of diagnosis. However, collection of a large series of uncommon tumours with homogeneous characteristics has become increasingly difficult, as prospective trials on patients with rare brain tumours are lacking. Currently, the choice of treatments and their timing are largely based on retrospective studies or small populations. Advances in the genomic, epigenomic, and transcriptomic fields have supplied greater information about molecular markers and actionable therapeutic targets, thereby improving diagnosis and tailoring treatment strategies [3].

In this paper, we aim to review the recent advances in targeting molecular pathways of rare brain tumours. We will discuss options such as targeting mTOR pathway in subependymal giant cells astrocytomas (SEGAs) of tuberous sclerosis and BRAF V600E mutation in rare glial (pleomorphic xanthoastrocytomas) or glioneuronal (gangliogliomas) tumours, which are a model of how specific molecular treatments can also favourably impact seizures and quality of life. Moreover, we will discuss initial experiences in targeting new molecular alterations in gliomas (IDH mutations and NTRK fusions) and in medulloblastomas (SHH pathway).

## 2. Review

### 2.1. Mechanistic Target of Rapamycin (mTOR)-Inhibitors in the Tuberous Sclerosis Complex

Tuberous sclerosis complex (TSC) is a genetic condition characterised by an increased risk of developing various types of benign tumours and hamartomas in multiple organs including the brain, skin, kidneys, heart, lungs, and eyes. Furthermore, TSC may be related to different neurological and psychiatric conditions such as epilepsy (often pharmacoresistant) and cognitive impairment. Familiar forms of TSC account for 30% of cases, while the others derive from de novo mutations of *TSC1* or -*2* genes [4]. The estimated incidence of TSC in recent studies ranges between 1/6000 and 1/10,000 live births. Thus, TSC truly represents a rare condition [5]. Typical brain lesions in patients with TSC are cortical tubers (non-evolving structural abnormalities of the cortex) and subependymal giant cell astrocytomas (SEGAs). SEGAs are slow growing tumours almost exclusively seen in TSC and are classified as grade I according to the 2016 WHO classification of brain tumours [6]. Even if SEGAs usually show an indolent behaviour, they become of clinical relevance as they often grow in the proximity of ventricles, obstruct the cerebrospinal fluid (CSF) pathways, and cause hydrocephalus and intracranial hypertension [7]. Furthermore, they are frequently diagnosed in the first two decades of life, thus impacting the quality of life of patients since childhood [8]. TSC is caused by loss of function of either *TSC1* or *TSC2*, encoding hamartin and tuberin, respectively. Hamartin and tuberin form a dimer which inhibits the protein Rheb (Ras homolog enhanced in the brain), an activator of mechanistic target of rapamycin (mTOR) complexes. mTOR consists of two distinct complexes with different cofactors. mTOR complex 1 (mTORC1) has the cofactor Raptor (regulatory-associated protein of Tor), while mTOR complex 2 (mTORC2) has the cofactor Rictor (rapamycin-insensitive component of Tor). mTORC1 is activated by Rheb [9]. Consequently, inactivation of *TSC1* or -*2* leads to hyperactivation of the mTORC1 pathway, which upregulates normal cortical development and cellular proliferation and induces the development of tubers and SEGAs [10]. Additionally, abnormal functioning of mTOR pathway plays a crucial role in TSC-related epilepsy [11,12]. Therefore, mTOR inhibitors can be effective against both tumour growth and control of seizures [13].

Everolimus is an mTOR inhibitor which is similar to rapamycin but has a higher affinity against the mTORC1 protein complex. Everolimus is widely employed as immunosuppressant agent to prevent rejection in organ transplantation. However, it is also known to be effective in TSC-related SEGA [14,15], angiomyolipoma [16], epilepsy [11,17,18,19], and neuropsychiatric disorders [20,21]. EXIST-1 was an international, prospective, double-blind, placebo-controlled phase 3 trial which explored the role of everolimus in patients with TSC-related SEGA [13,14]. Among 117 patients enrolled in the study, 78 and 39 were included in the everolimus and placebo groups, respectively. In the first analysis of the double-blind core phase [14], everolimus allowed 50% reduction in 27 (35%) patients, whereas no radiological or clinical response were seen in the placebo group (*p* < 0.0001). Furthermore, adverse events were mostly grade 1 or 2 and never determined treatment discontinuation. The most frequent adverse events (AEs) in the everolimus group were mouth ulceration (32%), stomatitis (31%), convulsion (23%), and pyrexia (22%). Due to these promising results, the study was prolonged with a long-term, open-label extension involving all patients remaining in the trial at the end of the core phase [15]. Among 111 patients who received one dose of everolimus, 57.7% achieved SEGA response (defined as at least 50% reduction in volume). Additionally, incidence of AEs was comparable with that of previous reports and occurrence of emergent AEs generally decreased over time. The most common AEs (30%) were stomatitis (43.2%) and mouth ulcerations (32.4%). Similar results were reported in the EFFECTS phase 3 trial, where partial responses accounted for 81 (67.5%) of patients, stable disease for 35 (29.2%), and progressive disease for 1 (0.8%) patient [22]. Finally, the EMINENTS trial aimed to evaluate the efficacy and safety of a reduced dose of everolimus (three times a week with a daily dose as in standard treatment-maintenance therapy) in a group of patients, who were previously treated with standard dose for at least 12 months. Stable disease was observed in 70.0% of patients, with a trend for tumour volume reduction. Interestingly, reduced maintenance dose of everolimus was not associated with major grade III-IV toxicity [23]. Table 1 reports the main studies investigating the role of everolimus in TSC-associated SEGA.

The role of everolimus in reducing seizure activity in TSC has been widely assessed. In a first prospective, multicentre, open-label phase 1/2 clinical trial, 20 patients with TSC and medically refractory epilepsy were treated with everolimus [11]. The median age of patients was eight years (range 2–21). Seizure frequency was reduced in 17 patients (85%), with a median reduction of 73% (*p* = 0.001). Additionally, a significant reduction in seizure duration and improvement in behaviour and quality of life were observed. At the end of the trial, 18 patients continued the treatment and 14 completed a second phase. A long-term analysis confirmed the positive results of the previous report, as at least 50% seizure reduction was maintained in 13 out of 14 patients [19]. Later, a randomised, double-blind, phase 3 trial (EXIST-3) explored the efficacy of everolimus on seizure control in a cohort of 366 children and adult patients (2–65 years) with TSC and drug-refractory epilepsy [17]. Placebo and two arms of everolimus with different dosage (low versus high dose) were compared. The response rate (≥50% seizure reduction) was 15.1% with the placebo, compared with 28.2% for low-dose everolimus (*p* = 0.0077), and 40.0% for high-dose everolimus (*p* < 0.0001). The median reduction in seizure frequency was 14.9% with placebo versus 29.3% with low-dose everolimus (*p* = 0.0028), and 39.6% with high-dose everolimus (*p* < 0.0001). Everolimus was well tolerated, stomatitis being the most frequent adverse event (up to 64% of cases). A discontinuation of treatment due to toxicity was limited to six patients in the low-dose group (5%) and four patients in the high-dose group (3%). Similar results were confirmed in a smaller EXIST-3-substudy on 35 Japanese patients and in a long-term analysis of the main trial focused on children and adolescents [18,21]. Finally, the efficacy of everolimus (4.5 mg/m^2^/day) in TSC-related neuropsychiatric disorders in TSC young patients (6–21 years) has been investigated in a prospective, double-blind randomised, placebo-controlled two-centre phase II study [20]. A comprehensive neurocognitive and behavioural examination was used at baseline, three months, and six months. However, no significant improvement in neurocognitive functioning or behaviour was seen in the everolimus group.

Table 2 reports the main studies investigating the role of everolimus in TSC-associated epilepsy.

### 2.2. BRAF-Inhibitors in BRAF-Mutant Brain Tumours

*BRAF*, alternately referred to as v-Raf murine sarcoma viral oncogene homolog B1, encodes for one of three members of the rapidly accelerating fibrosarcoma (RAF) serine/threonine kinase family. It is part of the mitogen activated kinase (MAPK) pathway, which enables cells to respond to extracellular growth signals (together with other similar pathways, such as Ras, Raf, MEK, and ERK). The activation of MAPK pathway fosters several intracellular signals which promote cell growth and survival. The pathological activation of such system (due to either BRAF gain of function or loss of function of regulatory proteins) is implicated in tumourigenesis [24]. In particular, gain of function of BRAF plays a critical role in a heterogeneous group of gliomas, as well as in the majority of metastatic melanoma and other cancers (including papillary thyroid cancer, colon carcinomas, hairy cell leukaemia) [25]. Gain of function of BRAF may result from either gene fusions or single nucleotide mutation. The most frequent fusion and single nucleotide mutation observed in brain tumours are KIAA1549:BRAF and BRAF V600E, respectively [26], even if a vast range of rarer alterations has been recognised [27].

BRAF alterations are a hallmark of different rare primary brain tumours. In a combined clinical and genetic institutional study of 510 patients with paediatric low-grade gliomas (PLGGs) [28], BRAF V600E mutation was detected in 69 of 405 patients (17%) and correlated with poor outcomes after chemotherapy and radiation therapy with a 10-year progression-free survival (PFS) of 27% (95% CI, 12.1% to 41.9%) and 60.2% (95% CI, 53.3% to 67.1%) for BRAF V600E-mutated and wild-type PLGG (*p* < 0.001), respectively. When exploring the prevalence of BRAF alterations in different subtypes, KIAA1549:BRAF fusion and BRAF V600E mutations were seen in 50–85% and 9-15% of pilocytic astrocytomas, respectively and were mutually exclusive [29,30]. BRAF single nucleotide mutations are observed in 63–70% of pleomorphic xanthoastrocytomas and 38% of the anaplastic variant. BRAF alterations may also be occasionally seen in diffuse lower-grade gliomas and glioblastoma, especially in the astrocytic lineage (up to 15% in grade II and III astrocytomas and 9% in glioblastomas). BRAF alterations are frequently seen in glioneuronal tumours: KIAA1549: BRAF fusion and BRAF V600E mutation are seen in 25% and 13–56% of gangliogliomas and gangliocytomas, respectively. BRAF single nucleotide mutations are found in 11%, 51%, 43% and 65% of desmoplastic infantile astrocytoma/ganglioglioma, dysembryoplastic neuroepithelial tumour, subependymal giant cell astrocytoma, and diffuse leptomeningeal glioneuronal tumour [24]. Growing attention has been devoted to design tailored therapies with MAPK-inhibitors in brain tumours with BRAF activation, with particular regard to paediatric low-grade gliomas (PLGGs) [26,31]. Dabrafenib, vemurafenib and encorafenib are small molecules targeting BRAF V600E mutation, while trametinib, selumetinib, and binimetinib are small molecules inhibiting MEK1/2 kinases. Their employ in primary brain tumours has been investigated in the following settings:Pilocytic astrocytoma. Combined BRAF and MEK inhibition has been proven to significantly improve the outcome of patients with BRAF V600E-mutant pilocytic astrocytoma in a small series of five paediatric cases [32]. Furthermore, in contrast to melanoma, monotherapy with MEK inhibitors may also be effective in pilocytic astrocytoma: in a phase 2 study of selumetinib on paediatric patients with pilocytic astrocytoma with either KIAA1549–BRAF fusion or BRAF V600E mutation, a sustained response rate was seen in 9/25 (36%) of patients, and prolonged stable disease in 11/25 (44%) [33]. Interestingly, both types of BRAF alterations were responsive to selumetinib, though the response rate was higher in tumours with BRAF fusions than in those with BRAF V600E mutation. Similarly, an ongoing clinical trial evaluating safety and efficacy of the MEK inhibitor binimetinib in children with BRAF-activated gliomas or other solid tumours is providing initial promising results [34].Paediatric diffuse gliomas. The use of BRAF-inhibitors in paediatric BRAF-activated grade 2 and 3 gliomas is giving encouraging results, as preliminarily demonstrated in studies involving small cohorts of patients [28,32,33,35,36,37,38]. In the aforementioned retrospective institutional study on 405 patients with PLGGs [28], 6 patients who experienced disease progression after conventional therapies were treated with BRAF inhibitors on a compassionate basis. All responded to targeted therapy, with no cases of tumour progression during treatment. In another institutional series of seven patients with BRAF V600E-mutant PLGGs treated with vemurafenib, only one patient progressed while on treatment, and an improvement of neurologic function was observed in all patients a few weeks after the start of therapy [37]. Moreover, in a first study exploring the efficacy of dabrafenib on 32 children (2–17 years) with refractory or progressing BRAF V600E-mutant, the response accounted for 41% of all patients [39]. Finally, selumetinib (a MEK inhibitor) is being studied in monotherapy in patients with LGG in a phase 1 study, and promising preliminary results have been published [35,36].Adult high-grade gliomas. Clinical experience about the use of MAPK-inhibitors in adult patients with BRAF-activated gliomas is limited and based on small cohorts of patients with recurrent tumours. Thus, the efficacy of BRAF-tailored therapies in adults is not as well established as in children. In a basket study of vemurafenib in adults with BRAF V600E-mutated gliomas, the response rate in GBM and anaplastic astrocytoma was 9% (1/11) only [40]. More encouraging results were provided by the interim analysis of a trial of combined dabrafenib/trametinib in adults with high-grade glioma, who displayed a response rate of 22% in grade 3 and 29% in grade IV glioma [41]. Additionally, a trial investigating the efficacy of combined therapy with another BRAF/MEK inhibitor combination (encorafenib/binimetinib) is currently ongoing (NCT03973918).Pleomorphic xanthoastrocytoma. There are many reports documenting cases of pleomorphic xanthoastrocytomas responding positively to targeted therapy [31]. In the same aforementioned basket trial enrolling adult patients with BRAF-activated gliomas [40], seven patients with pleomorphic xanthoastrocytomas were included. Of them, 42% responded to vemurafenib and another 42% had stable disease for more than six months.Ganglioglioma. Positive responses to targeted therapy with BRAF inhibitor monotherapy or combination therapy have been observed in both children and adults with refractory anaplastic ganglioglioma [31].

Table 3 reports the main studies investigating the role of f BRAF-inhibitors in BRAF-altered primary brain tumours.

The efficacy of treatments with MAPK-inhibitors are limited by the occurrence of drug-resistance over time, as previously seen in melanoma and other solid cancers [42]. Drug-resistance usually derives from upregulation of other pathway, increased activation or expression of surface receptor tyrosine kinases (RTKs), such as epidermal growth factor receptor (EGFR), or loss of feedback inhibition by extracellular signal-regulated kinase (ERK), that can lead to activation of RAF signalling, and possibly many other mechanisms [31]. Similarly, resistance to MAPK-inhibitors may occur in gliomas treated with targeted therapies. It is still unclear whether the pathophysiology underlying the onset of drug-resistance in gliomas is the same of that seen in other cancers. Finally, emerging evidence suggesting a potential role of BRAF activation in brain tumour-related epilepsy (BTRE) is acquiring increasing interest. In a recent metanalysis on 509 patients with BTRE, 193 had the BRAF V600E mutation (34%). As expected, BRAF mutation prevailed in patients with gangliogliomas. Furthermore, it was significantly associated with age at seizure onset [43]. To date, there are no available data investigating whether MAPK-inhibitors may affect seizure control in BRAF-activated brain tumours with BTRE.

### 2.3. IDH-Inhibitors in IDH-Mutant Diffuse Gliomas

Lower-grade gliomas include diffuse grade 2 and 3 tumours with an average incidence of less than 0.5 cases out of 100,000 people per year [1]. Mutations of the isocitrate dehydrogenase (*IDH*) 1 or 2 genes occur in up to 70% of lower-grade gliomas [44]. *IDH1/2* mutations correlate with some clinical and radiological characteristics, and better outcome after extended surgical resections and/or adjuvant radio-chemotherapy, as compared to the IDH wild-type counterpart [45]. Even if *IDH* status is not the only factor which determines the biological and clinical behaviour of glial tumours, the presence of the mutations has shaped the current classification of gliomas [6,46,47]. A huge body of evidence highlights the role of IDH mutations in the pathogenesis of diffuse gliomas (as well as other blood and solid cancers). First, IDH mutations are expressed uniformly in gliomas but not in normal cells, are early events, and remain relatively stable over time. Second, the product of the mutant enzyme D(-)-2-hydroxyglutarate (2-HG) is oncogenic by inhibiting α-ketoglutarate-dependent enzymes which are key regulators of the histone and DNA demethylation in cells. Ultimately, this results in a status of cellular hypermethylation which hinders cellular differentiation and alters the normal chromosomal topology with aberrant gene expression. Furthermore, the hypermethylation-based silencing of genes, which encode for immune checkpoint inhibitors (anti-PD-1 and PD-L1), may interfere with immune response [48,49,50]. Additionally, 2-HG mimics glutamate on NMDA receptors, thus increasing the risk of seizures in IDH-mutant lower-grade gliomas [51,52,53]. For all of these reasons, targeting IDH mutation may have a significant impact on the natural course of IDH-mutant gliomas.

Ivosidenib (AG-120) is a selective, potent inhibitor of the mutant IDH1 enzyme [54]. In both preclinical and human models, the administration of ivosidenib induced a significant reduction of intracellular 2-HG, thus corroborating the biological efficacy of the drug [55] and leading to a phase 1 clinical trial in acute myeloid leukaemia [56], a phase 3 trial in cholangiocarcinoma [57], and a phase 1 trial in chondrosarcoma [58], all with promising results. The use of ivosidenib has also been explored in a phase 1, open-label trial on 66 patients with IDH-mutant gliomas (32 with grade 2, 18 with grade 3, 12 with grade 4 tumours) [59]. All patients had a confirmed diagnosis of IDH1-mutant glioma, which was recurrent or not responding to initial surgery, radiation, or chemotherapy. The study comprised a dose escalation and a dose expansion phase. 500 mg once per day was chosen for the expansion cohort and was well tolerated with no significant toxicity apart from two cases (3%) of grade ≥3 adverse events, that were considered potentially treatment-related. In patients with nonenhancing glioma (*n* = 35), the objective response rate was 2.9%, with one partial response. Thirty of thirty-five patients (85.7%) with non-enhancing glioma achieved a stable disease compared with fourteen of thirty-one (45.2%) with enhancing glioma. Median progression-free survival was 13.6 months (95% CI, 9.2 to 33.2 months) and 1.4 months (95% CI, 1.0 to 1.9 months) for non-enhancing and enhancing cohorts, respectively. In conclusion, ivosidenib 500 mg once per day proved to be effective in non-enhancing *IDH*-mutant gliomas in reducing tumour growth, with a favourable safety profile.

Vorasidenib (AG-881) is a potent, oral, dual inhibitor of IDH1 and -2 mutations, and has been proven to penetrate the blood-brain barrier in several preclinical studies and inhibit 2-HG production in glioma tissue by more than 97% in an orthotopic glioma mouse model [60]. Furthermore, vorasidenib was evaluated in 76 patients with glioma in two phase 1 studies and was associated with a favourable safety profile at doses of <100 mg daily. Preliminary clinical activity was observed in non-enhancing gliomas in both studies, with an objective response rate (ORR) of 30.8% at 50 mg daily in a perioperative study and >90% 2-HG suppression at this dose level relative to untreated control samples [61]. Based on this rationale, vorasidenib at a dose of 50 mg daily is being investigated in a phase 3 trial against placebo in patients with *IDH*-mutant grade 2 gliomas, who have received surgery since at least one year and not more than five years. Eligible patients must present nonenhancing residual tumours, that do not need immediate radio- or chemotherapy (INDIGO trial) [62]. The primary endpoint of the trial is progression-free survival. Secondary endpoints include safety and tolerability, tumour growth rate assessed by volume, time to next intervention, overall survival, and quality of life. Due to the critical role of *IDH* mutations in tumour-related epilepsy, seizure activity, and neuro-cognitive functions will serve as exploratory endpoints.

### 2.4. Neurotrophic Tyrosine Receptor Kinase (NTRK) Inhibitors in NTRK-Activated Gliomas

Neurotrophic tyrosine receptor kinases (NTRKs) are a group of high-affinity receptors consisting of three families (NTRK1-2-3) with similar structures and intracellular signalling pathways. They are known to be involved in several cellular functions such as growth, differentiation, and apoptosis. When hyperactivated (usually due to aberrant fusions with other genes), NTRK fusions may play a role as oncogenic primers in several cancer settings [63]. Both paediatric and adult brain tumours may present NTRK fusions, which represent interesting targets.

NTRK fusions, which are usually rare events, prevail in a particular subset of non-brainstem high-grade gliomas in very young children (less than three years), where they may be found in up to 40% of cases (particularly TPM3-NTRK1 and ETV6-NTRK3 fusions) [64]. A significant prevalence of NTRK fusions has also been observed in pilocytic astrocytomas (about 15%) [65]. Conversely, in diffuse lower-grade gliomas, as well as in glioblastoma and diffuse intrinsic pontine gliomas, NTRK fusions are usually found in less than 2% of cases, while they were not observed in ependymomas or medulloblastomas [63].

Entrectinib (RXDX-101) is the first developed anti-NTRK fusions agent, which displays a secondary effect against ALK and ROS1 fusion proteins and is proven to penetrate the blood-brain barrier [66]. The efficacy in primary and secondary brain tumours has been assessed in phase 1 and 2 trials (ALKA-372-001, STARTRK-1 [67], STARTRK-2, and STARTRK-NG), with promising results. Furthermore, in a series of paediatric high-grade gliomas reported at 2019 American Society of Clinical Oncology (ASCO) meeting, all four patients achieved a radiological response, including a complete response [68].

Larotrectinib (LOXO-101) is highly specific for NTRK fusions [69] and has been investigated in several clinical trials on solid tumours of paediatric patients, including primary and secondary brain tumours (NCT02637687, NCT02122913, NCT02637687, and NCT02576431). In particular, nine patients with primary brain tumours were identified from NCT02637687 and NCT02576431 trials. Disease control was achieved in all patients. The best objective response to therapy was partial response in one patient (11%), whereas the other patients showed stable disease [70].

A second-generation of NTRK inhibitors includes repotrectinib-TPX-0005 and LOXO-195-BAY2731954, that are being explored in clinical trials in order to compare their efficacy with first-generation drugs and, more importantly, to tackle tumour resistance to first line compounds [71,72].

### 2.5. Sonic Hedgehog (SHH) Inhibitors in SHH-Activated Medulloblastomas

Medulloblastomas prevail in paediatric patients, with an incidence ranging from 0.48 cases out of 100,000 per year in children to 0.02 cases out of 100,000 per year in adults [1]. About 60% of adult medulloblastomas present SHH pathway activation, which correlates with an intermediate prognosis [73]. The SHH pathway participates in the expansion, migration, and differentiation of immature precursor cells from the external granule-cell layer to form the internal granule-cell layer during cerebellar maturation. A reactivation of the pathway may foster the development of medulloblastomas. Vismodegib is a ligand-specific inhibitor of the SHH pathway and has been identified as a potential drug against SHH-activated medulloblastomas [74]. Sonidegib is a similar compound blocking SHH [75]. A systemic review and metanalysis on five clinical phase 1/2 trials exploring the efficacy of anti-SHH agents vismodegib and sonidegib showed that the pooled objective response rate (ORR) of SHH-inhibitors was 37% in SHH-driven medulloblastomas. Sonidegib produced an ORR 1.87-fold higher than that of vismodegib (95%CI 1.23, 6.69). Among paediatric patients, the efficacy of sonidegib was 3.67-fold higher than that of vismodegib (*p* < 0.05). The rate of grade 3/4 drug-related toxicity was similar between patients receiving vismodegib and sonidegib (11.6% vs. 11.2%) [76]. Recently, a phase 1/2 trial comparing the association of vismodegib and temozolomide versus temozolomide alone in recurrent or refractory SHH-activated medulloblastoma failed to demonstrate a significant improvement of six months progression-free survival for the combined treatment [77].

## 3. Conclusions

Identification of actionable molecular targets is critical in rare brain tumours of adults and children, where both collection of retrospective series and clinical trials are commonly limited by low incidence rates. Molecularly-driven trial designs will be useful in order to identify effective targets in rare brain tumours. In this regard, basket trials represent the most optimal approach thus far. Several biological issues remain unsolved. First, the frequency of molecular alterations within different tumour histologies is still largely unknown. Second, the biological bases that explains the heterogeneity of responses to target agents among and across different tumour types have to be determined, as well as the potential heterogeneity of molecular targets over time. Furthermore, the ability of target agents to cross the blood-brain barrier, blood-tumour barrier and blood-CSF barrier must be defined. Additionally, some clinical issues still represent an unmet need. It will be crucial to define the optimal timing of treatment (early versus late) and duration of treatment in different tumour types. Secondary resistance to first-line agents should be properly prevented and managed with a combination of agents targeting different molecular pathways. Finally, it will be of utmost importance to explore new tools for monitoring responses. In addition to careful neurological examination and conventional MRI or PET, the possibility for detecting molecular pathways on neuroimaging (radiomics) and/or in CSF/blood (liquid biopsy) will represent a significant step forward.

## Figures and Tables

**Table 1 ijms-22-07949-t001:** Main studies investigating the role of mTOR inhibitor everolimus in subependymal giant cell astrocytomas (SEGAs) of tuberous sclerosis complex (TSC).

	Design	Patients (*n*)	Median Age	Endpoint(s)	Adverse Effects (ae)	Efficacy
Franz et al., 2016 EXIST-1 trial [15]	Phase III	111	9.5 years	Primary: SEGAs control rate (≥50% volume reduction) Secondary: time to SEGAs response/duration of SEGAs response Additional endpoints: angiomyolipoma/skin lesion response	Stomatitis (43.2%) Mouth ulceration (32.4%) Pneumonia (13.5%) Hypercholesterolemia (11.7%) Nasopharyngitis (10.8%) Pyrexia (10.8%) Grade 3 or 4 treatment-related AEs: 36.0%	Median duration of everolimus treatment: 47.1 months SEGAs control rate: 57.7% Median time to SEGAs response: 5.32 months Progression-free survival at 3 years after treatment initiation: 88.8%
Fogarasi et al., 2016 EFFECTS trial [22]	Phase III	120	11 years	Primary: safety evaluation Secondary: efficacy evaluation	Aphthous stomatitis (15.0%) Pyrexia (15.0%) Bronchitis (9.2%) Stomatitis (8.3%) Cough (5.0%) Diarrhoea (5.0%) Headache (5.0%) Mouth ulceration (5.0%) Sinusitis (5.0%) Grade 3 or 4 treatment-related AEs: 23.3%	Partial Response: 81 patients (67.5%) Stable Disease: 35 patients (29.2%) Progressive Disease: 1 patient (0.8%)
Bobeff, 2021 EMINENTS trial [23]	Single-centre, open-label, single-arm, prospective study	15	13.8 years	The efficacy and safety of a reduced dose of everolimus (three times a week with a daily dose as in standard treatment—maintenance therapy) in patients previously treated with standard dose for at least 12 months	Hypercholesterolemia (70.0%) Pharingitis (50.0%) Diarrhoea (30.0%) Anaemia (30.0%) Thrombocytopenia (30.0%) Bronchitis (20.0%) Hyperglicemia (10.0%) Neutropenia (10.0%) Liver enzymes increase (10.0%) No grade 3 or 4 toxicity with a maintenance therapy regimen	Stable disease in 70.0% of patients Trend for tumour reduction

**Table 2 ijms-22-07949-t002:** Main studies investigating the role of mTOR inhibitor everolimus in tuberous sclerosis complex (TSC)-related epilepsy.

	Design	Patients (*n*)	Median Age	Endpoint(s)	Efficacy
Krueger et al., 2013 [11]	Phase I–II	20	8 years	Primary: seizures reduction ≥50% Secondary: impact on electroencephalography (EEG), behaviour, quality of life	12/20 patients had seizure reduction ≥50%. In 17/20 cases, seizures had a median reduction of 73% as compared to baseline (*p* < 0.001). A long-term analysis [19] confirmed the previous results (13/14 patients maintained ≥50% seizure reduction)
French et al., 2016 EXIST-3 trial [17]	Phase III	366	10.1 years	Primary: seizure response rate in high vs. low everolimus exposure vs. placebo groups Secondary: seizure-free rate	Response rate (≥50% seizure reduction): 15.1% with placebo, 28.2% for low-dose everolimus (*p* = 0.0077), 40.0% for high-dose everolimus (*p* < 0.0001). Median reduction in seizure frequency: 14.9% with placebo versus 29.3% with low-dose everolimus (*p* = 0.0028), and 39.6% with high-dose everolimus (*p* < 0.0001). A long-term analysis on paediatric patients [18] and a Japanese substudy [21] confirmed the previous results.

**Table 3 ijms-22-07949-t003:** Main studies investigating the role of BRAF inhibitors in BRAF-altered primary brain tumours (predominantly harbouring BRAF V600E mutation).

	Design	Tumour	BRAF Alteration(s)	Drug	Patients (*n*)	Median Age	Endpoint(s)	Toxicity	Efficacy
Drobysheva et al., 2017 [32]	Case series	Pilocytic astrocytoma (PA)	BRAF V600E	Dabrafenib/dabrafenib + trametinib	2	Pt. 1: 5-year-old Pt. 2: 1-year-old	Tumour control rate with BRAF-inhibitors	Skin rash Fatigue	1 near complete response 1 stable disease
Lassaletta et al., 2017 [28]	Institutional series	Paediatric low-grade gliomas	BRAF V600E	BRAF-inhibitors (not specified)	6 (patients who progressed after first-line treatment)	/	Clinical and genetic data	/	A significant response to treatment (49% to 80% tumour volume reduction) in all patients
Banerjee et al., 2017 [35]	Phase 1	Paediatric low-grade gliomas	KIAA1549–BRAF fusion/BRAF V600E (presence of BRAF aberrations assessed in 19/38 patients)	Vemurafenib	38	13.3 years	Recommended dose, dose-limiting toxicities, pharmacokinetics, tumour BRAF aberrations, and treatment-related changes in tumour MRI.	Skin rash Diarrhea Creatine phosphokinase elevation and of selumetinib	Partial responses: 5/38 (13.1%)
Robison et al., 2018 [36]	Phase 1	Paediatric low-grade gliomas/other non haematological malignancies	Not specified	Binimetinib (MEK162)	19	9 years	maximum tolerated dose of MEK162	Creatine phosphokinase elevation (53%) Skin rash (47%) Lymphopenia (24%)	Progressive disease: 4 (3 in low-grade glioma patients) Partial response: 4. Data not provided for the remaining 11 patients.
Del Bufalo et al., 2018 [37]	Retrospective	Unresectable Paediatric low-grade gliomas (ganglioglioma, plemorphic xanthoastrocytoma, ganglioneurocytoma, pylocitic astrocytoma)	BRAF V600E	Vemurafenib	7	75.2 months	efficacy	Skin rash	Partial response: 3/7 Stable disease: 2/7 Complete response 1/7 Progressive disease: 1/7. Sustained response (CR, PR, SD) correlated with clinical improvement.
Kaley et al., 2018 [40]	Phase 2	Non-melanoma solid tumors and myeloma that harbours BRAFV600 mutations, including adult gliomas, such as malignant diffuse glioma (*n* = 11: six glioblastoma and five anaplastic astrocytoma), PXA (*n* = 7), anaplastic ganglioglioma (*n* = 3), pilocytic astrocytoma (*n* = 2), and high-grade glioma, not otherwise specified (*n* = 1)	BRAF V600E	Vemurafenib	24	32 years	response rate, progression-free survival, overall survival, and safety.	Arthralgia (67%) Melanocytic nevus (38%) Palmar-plantar erythrodysesthesia (38%) Photosensitivity reaction (38%)	Stable disease: 10/24 (41.7%) Partial response: 5/24 (20.8%) Progressive disease: 5/24 (20.8%) Complete response: 1/24 (4.2%) Not evaluable: 3/24 (12.5%) Median progression-free survival: 35 months
Wen et al., 2018 [41]	Phase 2	High-grade gliomas	BRAF V600E	Dabrafenib + trametinib	37	42 years	Tumour control rate	Fatigue (35%) Headache (30%) Rash (24%)	Partial response: 7/31 (22.6%) Complete response: 1/31 (3.2%) Median progression-free survival: 1.9 months Median overall survival: 1.7 months
Fangusaro et al., 2019 [33]	Phase 2	Pilocytic astrocytoma(group 1)NF1-related paediatric low-grade gliomas(group 2)	KIAA1549–BRAF fusion /BRAF V600E	Selumetinib	50(group 1: 25; group 2: 25)	Group 1: 9.2 years Group 2: 10.2 years	Primary: partial/complete response to selumetinib Secondary: progression-free survival (PFS); association between BRAF aberrations and PFS/response; MAPK aberration analysis by whole-exome/RNA sequencing; characterisation of inter-/intra-patient variability in pharmakocynetics of selumetinib.	Creatine phosphokinase elevation (68%) Hypoalbuminaemia (>60%) Skin rash (>50%) Anaemia (>50%) Gastric haemorrhage (>505) Liver enzymes increase (∼50%) The most frequent grade 3 or 4 adverse events were elevated CPK (5 [10%]) and maculopapular rash (5 [10%]	Group 1 Partial response: 9/25 (36%) Stable disease 9/25 (36%) Progressive disease 7/25 (28%) 2-years PFS: 70% No association between BRAF alterations (fusion vs. mutation) and treatment response/PFS. Group 2 Stable disease: 15/25 (60%) Partial response: 9/25 (36%) Progressive disease: 1/25 (4%) 2-years PFS: 96% No association between BRAF alterations (fusion vs. mutation) and treatment response/PFS.
Hargrave al, 2019 [38]	Phase 1/2	Paediatric low-grade gliomas	BRAF V600E	Dabrafenib	32	8.5 years	Safety, tolerability, and clinical activity	Fatigue (34%) Rash (31%) Dry skin (28%) Pyrexia (28%) Maculopapular rash (28%) Grade 3/4 AEs: 9 patients (28%)	Stable disease: 16/32 (50%) Partial response: 13/32 (41%) Progressive disease: 2/32 (6%) Complete response: 1/32 (3%) Median progression-free survival: 35 months

## Data Availability

Not applicable.

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
