# Peer review of "Targeted Therapies in Rare Brain Tumours"

_ijms, 2021, doi:10.3390/ijms22157949_

Round 1

Reviewer 1 Report

Targeted Therapies in Rare Brain Tumors

Comments to the Editor/Authors:

  1. Since the article heavily focuses on therapeutic agents with respect to rare central nervous system tumors, the topic of the manuscript and the information therein clearly fits within the scope of IJMS.
  2. Summary: The authors provide a review of the recent progress of therapeutics used toward rare central nervous system tumors. In particular, the authors discuss therapeutics being used against the mTOR pathway and BRAF V600E mutation in subependymal giant cell astrocytomas and glial tumors, respectively. In addition, the authors discuss inhibitors against the mutant isocitrate dehydrogenase enzyme, neurotrophic tyrosine receptor kinase fusions, and the sonic hedgehog pathway of certain tumors.
  3. There are numerous grammatical errors throughout the manuscript, which make the manuscript a bit difficult to read. I have highlighed some of these errors in the lines below.
  4. Line 13, Abstract. The acronym m-TOR should be defined at the outset. Also, insert the word “the” before m-TOR. Is it m-TOR or mTOR, as used in Line 72 and so forth.
  5. Line 17-18, Abstract. The acronyms IDH, NRK, and SHH should be defined.
  6. Line 22. The acronym CBTRUS should be defined.
  7. Line 24 and throughout the rest of the text. For numbers that represent “thousand”, please use a comma instead of a period. Also, as per significant figures, 23.79 can be rounded to 24. Thus, the sentence would read: “…approximately 24 out of 100,000 people per year…”
  8. Line 33. A period should replace the colon.
  9. Line 42. “thereby” should replace “thus”.
  10. Line 59. A hyphen is needed before “2”. That is the sentence would read: “…mutations of TSC1 and -2” Also, when referring to genes, the genes should be italicized.
  11. Line 60. 6000 should be “6,000”.
  12. Line 64. Insert words “according to” in lieu of “in”. The sentence would read “…according to the 2016 WHO classification…”
  13. Line 69. Are the authors referring to the gene or the protein? If the former, then it should be italicized. I believe they are referring to the gene, though the authors could be referring to the mRNA.
  14. Lines 75-78. Could the authors please confirm that inactivation of TSC1 and/or TSC2 upregulates only the mTOR1 pathway?
  15. Lines 76-77. The authors should further clarify the word “modulates”. If inactivation of TSC1 or -2 upregulates mTOR1 complex pathway, does this effect upregulate or downregulate cortical development and cellular proliferation…?
  16. Lines 69-78. Due to the writing, the interplay of the genes/proteins is a bit choppy, and thus make it somewhat difficult to visualize the pathway(s). It would help the manuscript if the authors could somehow rewrite these sentences (though the sentences are currently direct).
  17. The authors should either discuss the phase I-II trials by Krueger et al. or delete it from Table 1.
  18. The studies listed in Table 1 should be listed in the order that they are mentioned in the text.
  19. The additional studies discussed on Page 4 should be included in a Table.
  20. Line 137. BRAF should be italicized.
  21. Line 150. Could the authors please clarify the meaning of “…panorama is continuously evolving.”?
  22. Line 220. The table heading needs to be an actual heading.
  23. As Table 2 is almost exclusively pertaining to BRAF inhibitors for the BRAF V600E mutation, it could be essentially notated in Table 2 heading.
  24. For each of the BRAF inhibitors, the authors should provide a sentence about the chemical/molecular make-up of the inhibitors. For example, is an inhibitor a small molecule, an antibody, and etc.
  25. Lines 225-226. EGFR and ERK should be defined.
  26. Line 239. The gene IDH should be italicized.
  27. Line 278. “…penetrate the brain…” should be corrected to “…penetrate the blood-brain barrier…”.
  28. Lines 310-314. The acronyms should be defined.
  29. Line 342. The acronym DLT should be defined.
  30. Overall, additional descriptors or information pertaining to each of the inhibitors would help the manuscript, which would constitute only a minor revision of the manuscript.

Author Response

Dear reviewer 1,

thank you for your useful comments and suggestions.

Here, I provide a point-by-point response to your queries:

1-2. Nothing to comment.

3-13. Corrections done.

14-15. I found and corrected the following typo: TSC1/2 inhibit Rheb, which activates mTORC1 (I wrote accidentally the opposite in the submitted version of the manuscript). Thus, loss of function of TSC1/2 leads to hyperactivation of Rheb-mTORC1 pathway (which upregulates cell growth and proliferation).

16. I tried to describe the TSC1/2-Rheb-mTORC1 pathway in a clearer and more linear way.

17. I deleted the Phase I-II trial from the table, in order to focus to major and more recent phase III trials.

18. Correction done.

19. Table 2 summarises the results of the two major trials evaluating the role of everolimus in TSC-related epilepsy (a long-term extension and a Japanese substudy of the EXIST-3 trial have only been mentioned, as they mostly confirm the previous results of EXIST-3 study).

20. Correction done.

21. I deleted that confusing sentence.

22-23. Correction done.

24. See line 190.

25-29. Correction done

30. Mechanisms of action of everolimus (lines 85-89) and BRAF-inhibitors (lines 190-192) have been better described. Mechanisms of action of IDH, NTRK, and SHH-inhibitors  are reported in 2.3, 2.4, and 2.5 paragraphs, respectively. More detailed biopharmacological features of each inhibitor have been omitted in order to make the manuscript concise and more clinically-oriented. In particular, we tried to keep a broader focus on relevant trial findings and treatment recommendations.

Reviewer 2 Report

The author should make the article concise, highlighting the main therapies and trials for rare tumors. Also, the authors should thoroughly check for grammar and English.

Author Response

Dear Reviewer 2,

thank you for your comments.

We corrected several typos throughout the manuscript, which would make the article easier to read (see new manuscript attached below). Also, we added a second table focused on trials evaluating the employ of everolimus in TSC-related epilepsy and we better specified the mechanisms of action of each molecular inhibitors (see paragraphs 2.1-2.5). 

We hope that this last version of the manuscript will address your requests. 
